# Motivation in the Athens Classic Marathon: The Role of Sex, Age, and Performance Level in Greek Recreational Marathon Runners

**DOI:** 10.3390/ijerph16142549

**Published:** 2019-07-17

**Authors:** Pantelis T. Nikolaidis, Aïna Chalabaev, Thomas Rosemann, Beat Knechtle

**Affiliations:** 1Exercise Physiology Laboratory, 18450 Nikaia, Greece; 2School of Health and Caring Sciences, University of West Attica, 12243 Athens, Greece; 3UFR APS, Université Grenoble Alpes, 38400 Grenoble, France; 4Institute of Primary Care, University of Zurich, 8091 Zurich, Switzerland; 5Medbase St. Gallen Am Vadianplatz, 9001 St. Gallen, Switzerland

**Keywords:** gender, master athletes, endurance, marathon, personal goal achievement, physical motives, psychology

## Abstract

The aim of the present study was to examine the motivation of recreational runners and its variation by sex, age, and performance level. Finishers (female: *n* = 32, age 40.1 ± 9.0 years old, height 162 ± 7 cm, body mass 57.7 ± 7.5 kg, race record 4:34 ± 0:39 h:min; male: *n* = 134, 44.2 ± 8.6 years, 176 ± 6 cm, 77.0 ± 9.3 kg, 4:02 ± 0:44 h:min) in the Athens Classic Marathon 2017 completed the Motivations of Marathoners Scales (MOMS) 56-item questionnaire. The highest scores in the MOMS were observed in the general health orientation and personal goal achievement categories, and the lowest in the recognition and competition areas. Female participants scored higher in coping, self-esteem, and goal achievement than their male counterparts (*p* < 0.05). The <30 age group scored higher than the 35–40 and 40–45 age groups in “competing with other runners” for male participants (*p* < 0.05). The average performance group outscored the slowest group in “achieving personal goals” and “competing with other runners” in female participants, whereas an effect of performance on these two themes was shown in male participants as well (*p* < 0.05). In summary, we partially confirmed that female and male marathon runners differ for their motivations. In addition, novel findings were the identification of age and performance level as correlates of motivations. The knowledge of these trends would be of great practical value for practitioners to optimize the motivation of their athletes.

## 1. Introduction

Marathon running is one of the most popular endurance sports, with the number of races and finishers having increased dramatically during the last decades—e.g., participation in the New York City marathon increased by 119% from 1983 to 1999 [1]. An increased scientific interest has focused on physiological characteristics of marathon runners [2]; however, little information exists with regards to their psychological characteristics, such as motivation. Competition seems to be an important reason to run, at least more in marathon than in ultra-marathon runners [3]. Overall, the literature suggests the importance of taking into account runners’ motivations to better prevent injuries, considering the high prevalence of injuries in marathons (e.g., one-year prevalence of running injuries = ~55%) [4]. On the other hand, it has been observed that the highest motivations to run a marathon were intrinsic or task-related (meaning of life, self-esteem, and orientation towards health), whereas extrinsic or ego-related (recognition) motivations were the lowest [5]. More research is therefore needed to better identify marathon runners’ motivations.

In addition, runners’ motivations may vary according to sex, age, or performance level. So far, the focus has been placed on examining sex differences in motivation, probably due to the relatively low rates of female participation in marathons. In Spain, there is evidence that female runners seem to face the challenge of preparing for a marathon with more commitment and responsibility than male runners [6]. Female runners scored higher than male runners in four of the seven motivational sub-scales—self-esteem, achieving personal goals, affiliation with other runners, and concern about weight [5]. Moreover, female runners had higher scores than male runners in commitment, negative addiction to running, and all motivations [7].

Although the abovementioned research has started to investigate variations of motivations by sex, little information is available about the role of age and performance level. For instance, a study of motivations between two age groups (20–28 years versus >50 years) of male runners pre-registering for marathon races showed that older runners scored higher in general health orientation, weight concern, life meaning, and affiliation with other runners, and lower in personal goal achievement than younger runners [8]. Nevertheless, still there is a gap in the ~40 years age group, which is the mean age of marathon runners [9]. Knowledge about the effects of age and performance level on motivations would be of great practical interest for coaches working with marathon runners, as it is a common practice to train a small group of runners differing in age and performance level. It would be especially interesting to examine this topic in one of the most prestigious races, as it has been shown that the perception of the event prestige might influence the relationship between sport involvement and intent to return to the event [10]. Furthermore, motivation might differ by the performance level of athletes, as it has been observed that motivation varies by experience with experienced marathon runners, characterized more by competition, recognition, and health concerns than their less experienced counterparts [11]. Also, it has been found that finishing a marathon is related to psychological, physical health, and social motivations [12]. Therefore, the aim of the present study was to examine the effect of sex, age, and performance level on motivations in finishers of the Athens Classic Marathon 2017.

## 2. Materials and Methods 

### 2.1. Subjects

A total of 165 Greek recreational marathon runners (female: *n* = 32, age 40.1 ± 9.0 years old, height 162 ± 7 cm, body mass 57.7 ± 7.5 kg, race record 4:34 ± 0:39 h:min; male: *n* = 134, 44.2 ± 8.6 years, 176 ± 6 cm, 77.0 ± 9.3 kg, 4:02 ± 0:44 h:min), mostly from the area of Athens, volunteered to participate and completed all procedures. Female participants were classified into three age groups (<35 years, *n* = 9; 35–45 years, *n* = 13; >45 years, *n* = 10), whereas male participants were classified into eight groups (<30 years, *n* = 7; 30–35 years, *n* = 8; 35–40 years, *n* = 25; 40–45 years, *n* = 34; 45–50 years, *n* = 32; 50–55 years, *n* = 16; 55–60 years, *n* = 6; >60 years, *n* = 6). The classification of male participants into age groups was in agreement with typical age groups used in marathon races [13,14]. Female participants were also classified into three performance groups (fast: *n* = 10, <4:15 h:min; average: *n* = 11, 4:15–4:45 h:min; slow: *n* = 11, >4:45 h: min), and male participants into quartile performance groups (Q1, Q2, Q3, and Q4, Q1 being the fastest and Q4 being the slowest) based on their race time in the Athens Classic Marathon 2017. For comparison, 4:48 and 4:21 h:min were the mean race times in the New York City marathon from 2006 to 2016 for female and male finishers, respectively [9]. The number of age and performance groups differed between female and male participants, due to the higher number of male participants. 

### 2.2. Design

The present study is part of the Athens Classic Marathon project [2], which aims to profile the physiological and psychological characteristics of recreational marathon runners. For this purpose, the project was advertised through popular websites for endurance runners. During September and October 2017, the participants visited the laboratory, where they were examined for anthropometric and physiological characteristics, and completed questionnaires about training habits and motivation. The anthropometric and physiological characteristics of participants are available elsewhere [2,15,16]. All participants provided oral and written informed consent after having been informed in detail about the benefits and potential risks. All procedures were in accordance with the ethical principles for human experimentation derived by the Declaration of Helsinki and approved by the local institutional review board. 

### 2.3. Experimental Procedures

The methods, protocols, and equipment used in the assessment of anthropometric and physiological characteristics have been presented elsewhere [2,15,16]. Participants were instructed to abstain from exercise and intense physical activity for 24 h prior to their visit in the laboratory. After having been instructed in detail, they completed the 56-item Motivations of Marathoners Scales (MOMS), which has adequate internal consistency (Cronbach’s alpha range 0.80 to 0.93), retest reliability (intraclass correlation coefficient range 0.71 to 0.90), and factorial validity of the subscales [17]. It should be highlighted that MOMS was completed before performing anthropometric and physiological tests. Each item is scored using a seven-point Likert scale, where participants denote the degree of their agreement with each item—for example, “reason to run” ranging from 1 (it is not a reason to run) to 7 (it is a very important reason to run). MOMS has been used in many studies [18]. It identifies four broad categories or reasons for running (and nine specific themes within these categories): psychological (providing a sense of life meaning, enhancing self-esteem, and psychological coping), achievement (achieving personal goals and competing with other runners), social (desire to receive recognition and approval from others, and the desire to affiliate with other runners), and physical motives (general health orientation and concern about weight) [19].

### 2.4. Statistical Analysis

Descriptive statistics (mean ± standard deviation (SD)) were calculated for each of the nine MOMS specific themes. A one-way, repeated measures analysis of variance (ANOVA) and a Bonferroni post-hoc test compared scores among the nine specific themes. Cronbach’s alpha coefficients were calculated to test the internal consistency of these themes. An independent Student’s *t*-test examined the differences between female and male participants. A one-way ANOVA and a Bonferroni post-hoc test examined differences between age and performance groups in each sex. Statistical analyses were performed using IBM SPSS v.20.0 (SPSS, Chicago, IL, United States) and GraphPad Prism v. 7.0 (GraphPad Software, San Diego, CA, United States). The level of significance was set at α = 0.05. 

## 3. Results

The descriptive statistics and internal consistency of the nine MOMS specific themes can be seen in Table 1. The reliability of the specific themes was either excellent (psychological coping, affiliation, and recognition) or good (the six other themes). A moderate effect of sex was observed on coping (*d* = 0.65, *p* = 0.002), self-esteem (*d* = 0.63, *p* = 0.003), and personal goal achievement (*d* = 0.71, *p* = 0.001), with female participants scoring higher than male participants on these motives (Figure 1). No difference was observed among age groups in female participants (Figure 2). In male participants, a large main effect of age on “competing with other runners” was shown, where the <30 age group outscored the 35–40 and 40–45 age groups (*p* < 0.05, η^2^ = 0.158) (Figure 3).

In female participants, a large main effect of performance level on “achieving personal goals” and “competing with other runners” was observed (*p* < 0.05, η^2^ = 0.185–0.241), with the slowest group scoring lower than the average performance group (Figure 4). In male participants, a moderate main effect of performance level on “achieving personal goals” and “competing with other runners” was shown (*p* < 0.05, η^2^ = 0.063–0.071) (Table 2).

## 4. Discussion

The main findings of the present study were that (i) the highest scores in the MOMS were observed in the general health orientation and personal goal achievement, and the lowest in the recognition and competition; (ii) females scored higher male participants in coping, self-esteem, and goal achievement, but not in the other motives; (iii) the <30 age group outscored the 35–40 and 40–45 age groups in “competing with other runners” in male participants; and (iv) the average performance group outscored the slowest group in “achieving personal goals” and “competing with other runners” in both female and male participants.

Except for the “recognition” motive, scores in all themes were higher in the present study compared to the original research of Masters and colleagues, who used similar (~4) male-to-female participants’ ratio [17]. The increased motivations of participants in the Athens Classic Marathon might be partially attributed to the increased participation in marathon races during the last decades, indicating that this sport has become more popular and attractive. Another explanation of the increased motivation in the Athens Classic Marathon might be the prestige of the specific race, as it takes place in the same route run first by Pheidippides after the battle of Marathon, 490 BC [20]. Overall and in line with Ruiz-Juan and Sancho [5], this study indicates that recreational marathon runners are more motivated by personal reasons (health and personal goal achievement) than by competition or social recognition, which were the lowest motives reported by runners. This observation was in agreement with a general trend in the different motivations between recreational and professional athletes, where affiliation was important for the former group, in contrast to a strong achievement motivation for the latter group [21]. The observation of the highest scores in the general health orientation and personal goal achievement was partially in agreement with a study on female ultra-marathoners, which identified general health orientation and psychological coping as the two strongest motives [22]. Although both studies identified general health orientation as the most important motive, the score of female participants in the present study was higher than that of ultra-marathoners, highlighting the differences between marathoners and ultra-marathoners. For instance, it has been previously shown that marathoners outscored ultra-marathoners for competition motives [3].

The findings with regards to sex differences in motivations partially confirm previous observations [5,6,7,23]. Indeed, some sex differences were observed concerning psychological (self-esteem and psychological coping) and achievement (personal goal) motives. However, sex differences were observed in only three motives, and there were no significant differences between female and male participants on the other motives, related to life meaning, competing with others, social recognition and affiliation, and health and weight concerns. In sum, this study suggests that female and male recreational athletes run marathons for mostly the same reasons. A novel finding was the identification of an age effect, which was shown in male participants, where the <30 age group scored higher than the 35–40 and 40–45 age groups in “competing with other runners” An explanation of this observation might be that 35–40 and 40–45 age groups in male participants present the highest rates of finishers, and they are slower than their younger counterparts [9]. Despite being non-significant, we also recorded the highest score of health orientation and weight concern in the oldest age groups in both sexes. This trend was in agreement with results in ultra-marathon runners; in their case, as age increased, the concern for the physical benefits (i.e., health and weight) achieved through ultra-marathon running also increased [24].

Another novel finding was that the effect of performance level on “achieving personal goals” and “competing with other runners” in both sexes. It has been shown previously that elite football players were more motivated for performance than recreational athletes and non-athletes [25]. Thus, it seems that competitive marathon runners are more motivated than their recreational counterparts for performance, and this trend is similar with other sports.

### Limitations, Strength and Practical Applications

As the prestige of a race might influence the motivations of finishers [10], caution would be needed to generalize the findings from Athens Classic Marathon to races with less prestige. Also, other running race distances might vary for motivations. For instance, it has been observed that the most important factor motivating ultra-marathon runners was personal goal achievement [26], whereas mental well-being and socio-psychological aspects were important for half-marathon runners [27]. A further limitation of the present study was the different number of female and male participants, which resulted in unequal number of age and performance groups between female and male participants. Moreover, it should be noted that the male-to-female participants’ ratio (4.16) in the present study was very close to the male-to-female finishers’ ratio in Athens Classic Marathon 2017 (4.04) [14]. The strength of the study was its novelty, as it was the first one to examine the role of age and performance on motivations of marathon runners, in addition to sex. Considering the increased number of races and finishers in this sport, our findings are of great practical importance for coaches, fitness trainers, and sport psychologists working with runners, in order to optimize the motivation of their athletes considering sex, age, and performance level. Furthermore, data on the role of age and performance presented here might be used as reference in future studies. Strength and conditioning coaches usually work with marathon runners differing in sex, age, and performance level; thus, they should consider these aspects in order to optimize the motivation of their runners.

## 5. Conclusions

In summary, we partially confirmed that female and male marathon runners differed for their motivations. In addition, novel findings were the identification of age and performance level as correlates of motivations. The knowledge of these trends would be of great practical value for practitioners, in order to optimize the motivation of their athletes.

## Figures and Tables

**Figure 1 ijerph-16-02549-f001:**
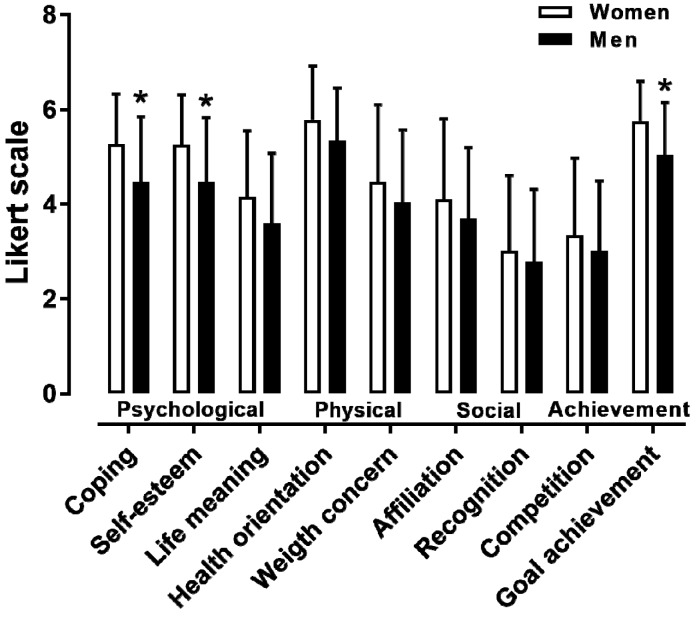
Nine MOMS-specific themes by sex. * *p* < 0.05. It should be noted that the Likert scale ranges from 1 to 7.

**Figure 2 ijerph-16-02549-f002:**
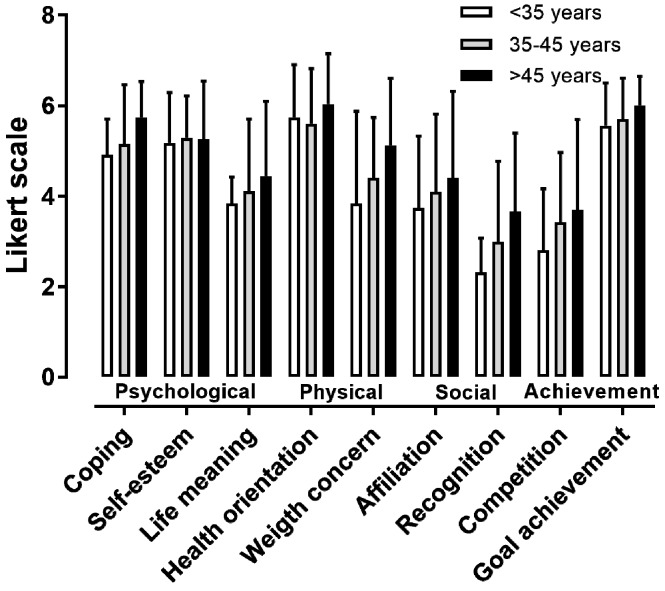
The nine MOMS specific themes by age group in female participants. It should be noted that the Likert scale ranges from 1 to 7.

**Figure 3 ijerph-16-02549-f003:**
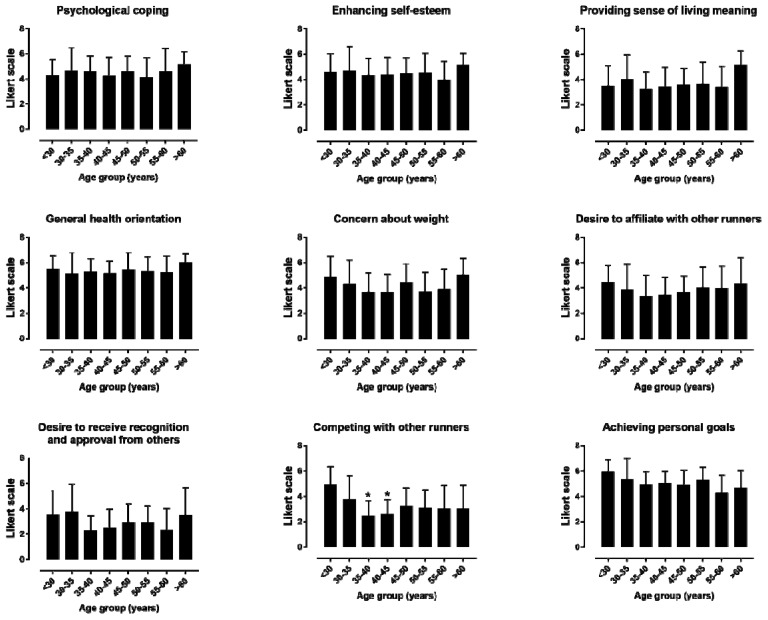
Nine MOMS specific themes by age group in male participants. * *p* < 0.05. It should be noted that the Likert scale ranges from 1 to 7.

**Figure 4 ijerph-16-02549-f004:**
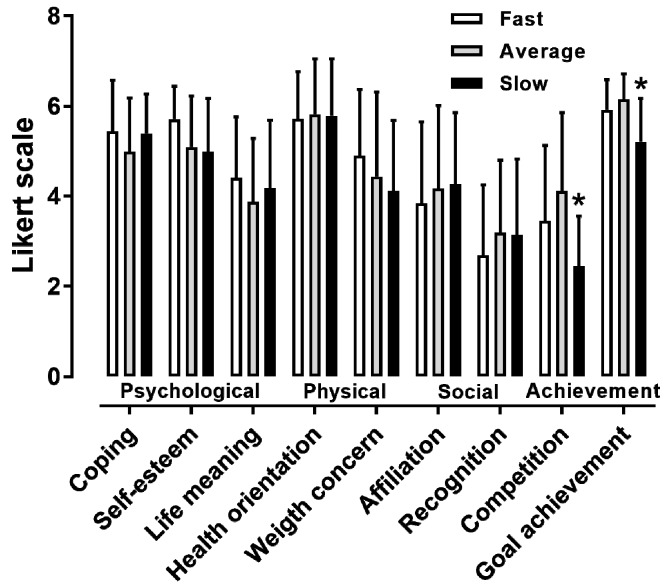
The nine MOMS specific themes by performance group in female participants. * *p* < 0.05. It should be noted that the Likert scale ranges from 1 to 7.

**Table 1 ijerph-16-02549-t001:** Descriptive statistics and internal consistency of the nine Motivations of Marathoners Scales (MOMS) specific themes.

Specific Theme	Mean ± SD	Range	Cronbach’s Alpha
*Psychological*			
Psychological coping	4.62 ± 1.36	1.00–6.89	0.904
Self-esteem	4.62 ± 1.34	1.13–7.00	0.879
Life meaning	3.69 ± 1.49	1.00–6.86	0.881
*Physical*			
General health orientation	5.43 ± 1.13	1.00–7.00	0.847
Weight concern	4.12 ± 1.56	1.00–7.00	0.843
*Social*			
Affiliation	3.78 ± 1.54	1.00–7.00	0.901
Recognition	2.83 ± 1.53	1.00–6.83	0.930
*Achievement*			
Competition	3.08 ± 1.51	1.00–7.00	0.858
Personal goal achievement	5.17 ± 1.10	1.67–7.00	0.797

**Table 2 ijerph-16-02549-t002:** The nine MOMS specific themes by performance group in male participants.

Specific Theme	Q1 (*n* = 32)	Q2 (*n* = 32)	Q3 (*n* = 35)	Q4 (*n* = 33)
*Psychological*				
Psychological coping	4.43 ± 1.36	4.14 ± 1.52	4.83 ± 1.17	4.40 ± 1.41
Self-esteem	4.55 ± 1.39	4.23 ± 1.45	4.66 ± 1.26	4.36 ± 1.33
Life meaning	3.64 ± 1.47	3.38 ± 1.51	3.62 ± 1.46	3.70 ± 1.61
*Physical*				
General health orientation	5.28 ± 1.07	5.52 ± 0.87	5.34 ± 1.16	5.27 ± 1.35
Weight concern	3.83 ± 1.55	4.19 ± 1.72	4.07 ± 1.53	4.05 ± 1.44
*Social*				
Affiliation	3.67 ± 1.62	3.50 ± 1.58	3.81 ± 1.45	3.83 ± 1.42
Recognition	2.96 ± 1.62	2.75 ± 1.60	2.62 ± 1.30	2.79 ± 1.58
*Achievement*				
Competition	3.60 ± 1.59	3.13 ± 1.49	2.72 ± 1.15	2.74 ± 1.55
Personal goal achievement	5.40 ± 1.10	4.81 ± 1.19	5.25 ± 0.94	4.68 ± 1.12

Q1, Q2, Q3, and Q4 are performance groups based on race time quartiles, with Q1 being the fastest and Q4 the slowest. Data are presented as mean ± standard deviation (SD).

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
