# Peer review of "Motivation in the Athens Classic Marathon: The Role of Sex, Age, and Performance Level in Greek Recreational Marathon Runners"

_ijerph, 2019, doi:10.3390/ijerph16142549_

Round 1

Reviewer 1 Report

There is a gret diffenrence in numbers of male and female runners. Include in discussion the consequences for analysis of the data such as age effects and performance level

Author Response

Comments and Suggestions for Authors

There is a great difference in numbers of male and female runners. Include in discussion the consequences for analysis of the data such as age effects and performance level

Answer: We agree with the expert reviewer and added this aspect of the different number of male and female runners in the discussion (“A further limitation of the present study was the different number of female and male participants which resulted in unequal number of age and performance groups between female and male participants. Moreover, it should be highlighted that the male-to-female participants’ ratio (4.16) in the present study was very close to the male-to-female finishers’ ratio in Athens Classic Marathon 2017 (4.04) [14].”).

Reviewer 2 Report

This work looks good - it is valuable indeed, I like it a lot.

However, it should be cleaned/corrected a bit and, especially, enriched a bit --- especially by other or possible further research methodologies. There must be some fresh new formal aspects also.

Please revise the paper with focus and readiness to make in excellent in both contents (and informal style). In fact, I am very positive - however, it should more "powerful". I expect and trust that the authors will achieve this goal.

I recommend a Modest Revision --- between Major and Minor --- but seen as a chance (to widen it - open it to future applications, even in other sports and other areas of life ("Generalization Effect")).

Presently, main "drawbacks" are:

--- The authors should think a bit "wider", more addressing flaws of their concept and make the paper somewhat having a wider view; especially, there should be

(a) commented more recent and interesting other research and modelling techniques (example https://doi.org/10.1016/j.psychsport.2018.04.004 );

(b) other/future concepts of marathon runners motivation management:

(c) some further future research directions,

Please look for further works by the teams of Joan CarlesTrullàs, Matthew Lamont, Millicent Kennelly, Noel Brick, Tadhg MacIntyre, Mark Campbell etc. Part and the References could be extended accordingly.

Please, make wider and specific making conclusions and make recommendations.

Please check references and adopted according to requirements.

In the article no information about ethical Principles and Guidelines for the Protection of Human Subjects of the Research.

Example source nr 19 no journal name (Acta Facultatis Educationis Physicae Universitatis Comenianae).

Eventually.

You should very carefully revisit it in terms of other approaches in literature and new pathways of future research.

Please do it with interest and care.

Then the works could become acceptable.

Best wishes!

Author Response

Comments and Suggestions for Authors

This work looks good - it is valuable indeed, I like it a lot.

However, it should be cleaned/corrected a bit and, especially, enriched a bit --- especially by other or possible further research methodologies. There must be some fresh new formal aspects also.

Answer: We agree with the expert reviewer and added new aspects in the discussion.

Please revise the paper with focus and readiness to make in excellent in both contents (and informal style). In fact, I am very positive - however, it should more "powerful". I expect and trust that the authors will achieve this goal.

Answer: We agree with the expert reviewer and improved the content.

I recommend a Modest Revision --- between Major and Minor --- but seen as a chance (to widen it - open it to future applications, even in other sports and other areas of life ("Generalization Effect")).

Answer: We agree with the expert reviewer and added practical applications.

Presently, main "drawbacks" are:

--- The authors should think a bit "wider", more addressing flaws of their concept and make the paper somewhat having a wider view; especially, there should be

Answer: We agree with the expert reviewer and attempted to make it “wider” as suggested.

(a) commented more recent and interesting other research and modelling techniques (example https://doi.org/10.1016/j.psychsport.2018.04.004 );

Answer: We agree with the expert reviewer and added more relevant literature as suggested (“Also, other running race distances might vary for motivations. For instance, it has been observed that the most important factor motivating ultra-marathon runners was personal goal achievement [26].” in the 4.1. section).

(b) other/future concepts of marathon runners motivation management:

Answer: We agree with the expert reviewer and developed future directions for this topic.

(c) some further future research directions,

Please look for further works by the teams of Joan CarlesTrullàs, Matthew Lamont, Millicent Kennelly, Noel Brick, Tadhg MacIntyre, Mark Campbell etc. Part and the References could be extended accordingly.

Answer: We agree with the expert reviewer and added relevant literature and discussion.

Please, make wider and specific making conclusions and make recommendations.

Answer: We agree with the expert reviewer and revised the conclusions as suggested.

Please check references and adopted according to requirements.

Answer: We agree with the expert reviewer and revised all references carefully.

In the article no information about ethical Principles and Guidelines for the Protection of Human Subjects of the Research.

Answer: We agree with the expert reviewer and added this information in the Methods (“the ethical principles for human experimentation derived by”).

Example source nr 19 no journal name (Acta Facultatis Educationis Physicae Universitatis Comenianae).

Answer: We agree with the expert reviewer and corrected it.

Eventually.

You should very carefully revisit it in terms of other approaches in literature and new pathways of future research.

Answer: We agree with the expert reviewer and developed these aspects.

Please do it with interest and care.

Then the works could become acceptable.

Best wishes!

Reviewer 3 Report

The following study sought to describe motivation levels during marathon running and how sex, age, and performance level may impact these measures in recreational runners. The focus of the study is good and will be of interest to readers. Overall, the manuscript is well written and the authors are commended for successful completion of the manuscript. Below are some comments and suggestions which may improve the merit of manuscript:

Line 13: change “women” to “females”. This should be done all throughout the manuscript.

Line 14: Same thing for “men” to “males”.

Line 37: change “later” to “latter”

Line 37: Did this study measure motivation or any psychological variables? How is this study relevant?

Line 47: Break this sentence up. As written it is too long.

Line 54: change “was” to “is”

Line 55: What did they find? Authors should discuss their results to educate the reader further.

Overall, the introduction needs more background information on motivation and performance level. See studies below which may be relevant. Experience and training level would be good background to set up the novelty of your analysis based on performance.

Masters, Kevin S., and Benjamin M. Ogles. "An investigation of the different motivations of marathon runners with varying degrees of experience." Journal of Sport Behavior 18.1 (1995): 69.

Newcomer, Belinda Dawn. Motives of marathon runners in training: Investigating the differences between gender, experience level and age. Capella University, 2008.

Line 67-75: Where there any circumstances/criteria in which participants were excluded?

Line 70: Where did these age ranges come from? Source? What is the justification for these?

Line 71: How many participants were in each age range?

Line 118: The gray bars in figure 2 are very difficult to see. Authors should add black outline to the bars denoting the mean and change the error bar color to black. This should be fixed throughout the manuscript.

Line 125: How were the female times separated into “fast” “slow” and “average”? What criteria was used to categorize? Is there a source or rationale for this separation?

Line 152: how does this compare to other recreational athletes?

Line 161: change “confirmed” to “confirm”

Line 168: I do not know that “outscored” is the best word for likert scale findings since it is ordinal data. I think stating that participants “agreed more” or something similar may be better.

Line 181-190: The most glaring limitation of this study is the disparity between the number of males to females. Thus, that only allowed for three different age groups of females and findings may be different with a larger sample size. The authors have done a good job explaining the reason for the differences in numbers, but it still should be mentioned in the limitations section.

Line 182: The strength…

Author Response

Comments and Suggestions for Authors

The following study sought to describe motivation levels during marathon running and how sex, age, and performance level may impact these measures in recreational runners. The focus of the study is good and will be of interest to readers. Overall, the manuscript is well written and the authors are commended for successful completion of the manuscript. Below are some comments and suggestions which may improve the merit of manuscript:

Line 13: change “women” to “females”. This should be done all throughout the manuscript.

Answer: We agree with the expert reviewer and corrected it accordingly.

Line 14: Same thing for “men” to “males”.

Answer: We agree with the expert reviewer and corrected it accordingly.

Line 37: change “later” to “latter”

Answer: We agree with the expert reviewer; however, we deleted the whole sentence (see the next comment).

Line 37: Did this study measure motivation or any psychological variables? How is this study relevant?

Answer: We agree with the expert reviewer and deleted this sentence.

Line 47: Break this sentence up. As written it is too long.

Answer: We agree with the expert reviewer and broke it into three sentences.

Line 54: change “was” to “is”

Answer: We agree with the expert reviewer and corrected it accordingly.

Line 55: What did they find? Authors should discuss their results to educate the reader further.

Answer: We agree with the expert reviewer and added this information (“showed that older runners scored higher in general health orientation, weight concern, life meaning and affiliation with other runners, and lower in personal goal achievement than younger runners”).

Overall, the introduction needs more background information on motivation and performance level. See studies below which may be relevant. Experience and training level would be good background to set up the novelty of your analysis based on performance.

Masters, Kevin S., and Benjamin M. Ogles. "An investigation of the different motivations of marathon runners with varying degrees of experience." Journal of Sport Behavior 18.1 (1995): 69.

Newcomer, Belinda Dawn. Motives of marathon runners in training: Investigating the differences between gender, experience level and age. Capella University, 2008.

Answer: We agree with the expert reviewer and enhanced the introduction section and discussed the recommended literature (“Furthermore, motivation might differ by performance level of athletes as it has been observed that motivation varied by experience with experienced marathon runners characterized more by competition, recognition and health concern than their less experience counterparts [11]. Also, it has been found that marathon finishes were related with psychological, physical health and social motivations [12].”).

Line 67-75: Where there any circumstances/criteria in which participants were excluded?

Answer: We agree with the expert reviewer and added this information.

Line 70: Where did these age ranges come from? Source? What is the justification for these?

Answer: We agree with the expert reviewer and added this information.

Line 71: How many participants were in each age range?

Answer: We agree with the expert reviewer and added the number of participants by group.

Line 118: The gray bars in figure 2 are very difficult to see. Authors should add black outline to the bars denoting the mean and change the error bar color to black. This should be fixed throughout the manuscript.

Answer: We agree with the expert reviewer and revised the figures 2 and 4 accordingly.

Line 125: How were the female times separated into “fast” “slow” and “average”? What criteria was used to categorize? Is there a source or rationale for this separation?

Answer: We agree with the expert reviewer and added this information (“(fast, n=10, <4:15 h:min; average, n=11, 4:15-4:45 h:min; slow, n=11, >4:45 h:min). For comparison, 4:48 h:min was the mean race time in the New York City marathon from 2006 to 2016”).

Line 152: how does this compare to other recreational athletes?

Answer: We agree with the expert reviewer and added this aspect (“This observation was in agreement with a general trend in the different motivations between recreational and professional athletes, where affiliation was important for the former group, in contrast to a strong achievement motivation for the latter group [16].”).

Line 161: change “confirmed” to “confirm”

Answer: We agree with the expert reviewer and changed it accordingly.

Line 168: I do not know that “outscored” is the best word for likert scale findings since it is ordinal data. I think stating that participants “agreed more” or something similar may be better.

Answer: We agree with the expert reviewer and revised this terminology.

Line 181-190: The most glaring limitation of this study is the disparity between the number of males to females. Thus, that only allowed for three different age groups of females and findings may be different with a larger sample size. The authors have done a good job explaining the reason for the differences in numbers, but it still should be mentioned in the limitations section.

Answer: We agree with the expert reviewer and developed this aspect in the limitations section (“A further limitation of the present study was the different number of female and male participants which resulted in unequal number of age and performance groups between female and male participants. Furthermore, it should be highlighted that the male-to-female participants’ ratio (4.16) in the present study was very close to the male-to-female finishers’ ratio in Athens Classic Marathon 2017 (4.04) [20].”).

Line 182: The strength…

Answer: We agree with the expert reviewer and corrected it accordingly.

Reviewer 4 Report

This is a nice written study aimed to examine the effects of sex, age and performance level on motivations in finishers of the 2017 Athens Classic Marathon. I have some minor suggestions for improvement of this paper:

1) L80: the description of anthropometric measurements is missing in the ‘Methods’ section;

2) L81: I did not find any physiological characteristics of runners in this study;

3) Paragraph 2.3. needs more details, e.g. were runners instructed on how to fill out the questionnaires? What about the mood, fatigue during completing the questionnaires? Were they instructed how to prepare for visiting the lab, e.g. not after the training, or the time of the day?

5) Please, provide the number of participants in particular groups.

6) Please, emphasize in text and title that you involved only Greek runners in the project.

Author Response

Comments and Suggestions for Authors

This is a nice written study aimed to examine the effects of sex, age and performance level on motivations in finishers of the 2017 Athens Classic Marathon. I have some minor suggestions for improvement of this paper:

1) L80: the description of anthropometric measurements is missing in the ‘Methods’ section;

Answer: We agree with the expert reviewer and added this aspect in the Methods (“Methods, protocols and equipments used in the assessment of anthropometric and physiological characteristics have been presented elsewhere [2,15,16].”).

2) L81: I did not find any physiological characteristics of runners in this study;

Answer: We agree with the expert reviewer and provided references to the studies focusing on physiological characteristics (“The anthropometric and physiological characteristics of participants are available elsewhere [2,15,16].”).

3) Paragraph 2.3. needs more details, e.g. were runners instructed on how to fill out the questionnaires? What about the mood, fatigue during completing the questionnaires? Were they instructed how to prepare for visiting the lab, e.g. not after the training, or the time of the day?

Answer: We agree with the expert reviewer and these details were added (“Participants had been instructed to abstain from exercise and intense physical activity for 24h prior to their visit in the laboratory. After having been instructed in details, they...”; “It should be highlighted that MOMS was completed before performing anthropometric and physiological tests”).

5) Please, provide the number of participants in particular groups.

Answer: We agree with the expert reviewer and added the number of participants by group.

6) Please, emphasize in text and title that you involved only Greek runners in the project.

Answer: We agree with the expert reviewer and highlighted this detail in both title (“…in Greek recreational marathon runners”) and methods.